# Treatment with Oral Ondansetron for Ultramarathon-Associated Nausea: The TOO FUN Study

**DOI:** 10.3390/sports9030035

**Published:** 2021-03-03

**Authors:** Andrew V. Pasternak, David Fiore, Arthur Islas, Sarah Toti, Martin D. Hoffman

**Affiliations:** 1Silver Sage Center for Family Medicine, Reno, NV 89521, USA; 2Department of Family and Community Medicine, Reno School of Medicine, University of Nevada, Reno, NV 89557, USA; dfiore@med.unr.edu (D.F.); aislas@med.unr.edu (A.I.); 3Reno School of Medicine, University of Nevada, Reno, NV 89557, USA; stoti@med.unr.edu; 4Physical Medicine and Rehabilitation Service, Department of Veterans Affairs, Northern California Health Care System, Sacramento, CA 95655, USA; mdhoffman@ucdavis.edu; 5Department of Physical Medicine and Rehabilitation, University of California Davis, Sacramento, CA 95616, USA

**Keywords:** running, antiemetic, serotonin 5-HT3 receptor antagonist, endurance exercise, randomized controlled trial

## Abstract

Nausea and vomiting are common for runners during ultramarathons and often contribute to non-finishes. We aimed to determine the efficacy of ondansetron, a commonly used antiemetic, to treat nausea and vomiting in runners during an ultramarathon. Runners who had a previous history of frequent nausea or vomiting during races and entered in 160, 80, and 55 km ultramarathons in 2018 and 2019 were randomized in a double-blind fashion to 4 mg ondansetron or placebo capsules to use if they developed nausea or vomiting during the race with the ability to take three additional doses. Study participants completed a post-race online survey to assess medication use and efficacy. Of 62 study participants, 31 took either ondansetron (20) or placebo (11). In this small study, there were no group differences in those reporting any improvement in nausea and vomiting (*p* = 0.26) or in the amount of improvement (*p* = 0.15). We found no evidence that ondansetron capsules improve nausea and vomiting during ultramarathons.

## 1. Introduction

Athletes running in ultramarathon events commonly have upper gastrointestinal (GI) distress, with nausea present in up to 60% [1]. Nausea or vomiting is the primary reason for not finishing a 161-km ultramarathon, and even among those completing the race, over a third reported, it impacted their race performance [2]. 

During exercise, GI distress has a wide range of etiologies, including dehydration, heat stress, catecholamine secretion, use of non-steroidal anti-inflammatories, altitude exposure, hyponatremia, competition anxiety, and the amount and content of food and beverage consumed [3,4,5]. In one study of runners during an ultramarathon [6], those who ingested a higher percentage of calories from fat and drank more fluids had less nausea and vomiting, perhaps due in part to improving the distribution of blood flow to the GI tract [7]. 

Treating upper GI distress in runners during an ultramarathon is challenging. Ondansetron, commonly used to treat nausea and vomiting, is a serotonin 5-HT3 receptor antagonist. It is used in various settings such as chemotherapy, gastroenteritis, and postoperative nausea and vomiting [8,9]. Anecdotally, ondansetron may help exercise-related nausea and vomiting; however, there has been no randomized control trial to support this claim [10]. In a small unblinded pilot study at the Tahoe Rim Trail Endurance Races (TRTER), 66% of 21 runners felt their symptoms improved after taking a 4 mg orally dissolvable ondansetron tablet sublingually [11]. One of the concerns about using ondansetron in ultramarathons is a theoretical risk of it contributing to heat-related illnesses. Ondansetron, however, has not been shown to affect thermoregulation in hot settings [12]. 

As the etiology of nausea and vomiting in endurance athletes may differ from nausea and vomiting in other settings, we felt that the pilot study’s findings warranted further exploration. We have been unable to find other studies examining the effectiveness of ondansetron in this setting. Thus, the purpose of this double-blind, randomized control trial was to evaluate the effectiveness of ondansetron at treating ultramarathon-associated nausea or vomiting. 

## 2. Materials and Methods

Pre-race recruitment and randomization: We recruited runners from the 2018 and 2019 TRTER. The TRTER has over 600 runners competing in 55, 80, and 160 km races, with approximately 200 runners in each event. Pre-race emails, sent to all competitors, invited them to consider enrolling in the study if they had nausea or vomiting during previous races. Before enrollment, we screened study participants for previous reactions to ondansetron or medications that could interact with ondansetron. Weather conditions, aid station food choices, and sports drinks supplied at the race were similar between years. 

Study participants were randomized using a random number generator to either ondansetron 4 mg oral capsules or placebo (cellulose capsule) [13]. A local compounding pharmacy prepared the capsules. Each runner was given four capsules to carry with them during the race with instructions to take one capsule if they developed symptoms. They could then take an additional capsule every 4 h as needed based on recommended dosing. A total of 62 runners elected to participate in the study. Of these, the average age was 45 years old (range 21–78). With respect to gender, 25 were women, and 37 were men.

Race day logistics: Runners encounter medical staff two, four, and nine times in the 55, 80, and 160 km races, respectively. The medical staff were blinded to group allocation, but if a study participant presented with persistent GI symptoms after taking a study capsule, the medical staff could break the blinding to allow for additional treatment options. We asked the runners to record their distance and the time of day or elapsed race time when taking capsules.

Post-Race evaluation: Within 24 h of the race finish, emails were sent to study participants with a link to an online post-race survey, with follow up emails at three and seven days if needed The post-race survey involved 26 questions (available as Appendix A). Included were questions inquiring about previous experiences with nausea and vomiting and their racing history. Runners were asked if they felt the treatment helped (Yes/No/Not sure). They were also asked about the severity of symptoms and improvement of symptoms after treatment on a 1 to 100-point scale. For medication effectiveness, 1 = no help at all and 100 = complete resolution of symptoms. For symptom severity, 1 = very mild symptoms and 100 = severe symptoms. They were also asked to rate the severity of their nausea and vomiting on the same scale before taking the medication, one hour after taking the medication, and at the end of the race. 

Assuming a placebo effect of 30% and a 30% greater effect of ondansetron (60% effectiveness) as clinically significant, we would have needed 28 runners in each group to show a statistical effect. We chose 60% effectiveness of ondansetron based on the amount of improvement seen in our pilot study [11]. 

Group comparisons of ordinal data were made with two-tailed unpaired t-tests or the Mann–Whitney test when data were found to be skewed by the D’Agostino and Pearson omnibus normality test. Group comparisons of categorical data were made with the Fisher’s exact test. 

## 3. Results

All 62 runners completed the post-race survey within seven days, and 36 completed it within three days of the race finish. Of the runners who enrolled in the study, 31 (50%) had GI symptoms prompting them to take the treatment. Of those 31 runners, 20 were in the ondansetron arm, and 11 were in the placebo arm. Details on the number of study participants and race distances are in Figure 1


As shown in Table 1, the two groups were comparable in individual and during race characteristics, including the proportion that were unblinded during the race. Those taking ondansetron received no benefit compared with placebo. While not reaching significance, there were trends towards the placebo group having more improvement and more side effects than the ondansetron group. On average, 160 km runners took the treatment during the second quartile of the race by distance when ambient temperatures were highest. Weather both years was relatively mild and relatively similar; in 2018, temperatures ranged from 12 to 25 °C, and in 2019, temperatures ranged from 10 to 23 °C near the start/finish line. 

## 4. Discussion

Ondansetron did not improve participants’ symptoms of nausea and vomiting during an ultramarathon. Although it did not reach statistical significance, the placebo group tended to have more benefit and more side effects than the ondansetron group, suggesting that a larger subject group was unlikely to demonstrate a benefit of ondansetron. The side effects due to ondansetron were low, and there was no pattern for particular side effects.

The trend of the placebo group towards having greater benefit and more side effects is likely a reflection of low statistical power. However, it is interesting to see such a high placebo effect. Placebos have been shown to have an equal benefit to active medications, and up to 26% of people will have side effects from placebos [14]. In studies on postoperative nausea and vomiting, while aromatherapy treatment is not statistically superior to placebo, symptom scores often improve in both groups [15]. Perhaps, given the nature of nausea and vomiting during endurance events, athletes might be even more susceptible to a similar placebo/nocebo effect. 

Study power was limited, as only 31 (50%) of the study participants had symptoms prompting them to take the treatment. The limited numbers also made the subgroup comparison difficult to interpret. Based on previous studies showing high GI distress rates, we anticipated a higher percentage would try the treatment [1,2,11]. Some subjects may have been concerned about trying a new medication during a race. It may have been beneficial to provide sample ondansetron to study participants to try during training in advance of the study. We report the data before reaching the desired sample size because there was no suggestion of potential benefit from the active treatment.

Another limitation was that we asked runners retrospectively about their nausea and vomiting. We chose this approach as it would have been logistically challenging to capture these data before runners took the medication during the race as the racecourse covers 80 km, and the investigators had limited access to the runners. Except for the few runners who were unblinded during the race, runners were still blinded to their study group when they filled out the survey, which helped reduce response bias. 

Our study used ondansetron capsules that require gastric absorption instead of orally dissolvable tablets, which allow for buccal absorption. It is certainly possible that runners given ondansetron were not able to absorb the medication due to vomiting. During exercise, as blood is shunted from the GI tract to muscle, absorption of medications in the stomach may be limited. We made this choice as we were unable to find orally dissolvable placebo tablets that could be packaged to prevent breaking apart during an ultramarathon. In our non-blinded pilot study using orally dissolvable tablets, we saw more improvement in nausea and vomiting; this study does not exclude the possibility of buccally absorbed ondansetron effectiveness. It may have also been beneficial to use an 8 mg dose, which is commonly used in clinical practice, as opposed to a 4 mg dose [13]. 

In summary, while ondansetron did not show benefit, the study was limited due to the number of runners who chose to take the intervention and perhaps by the form and dose of ondansetron used. Further research on the use of ondansetron should consider these issues in study design. 

## Figures and Tables

**Figure 1 sports-09-00035-f001:**
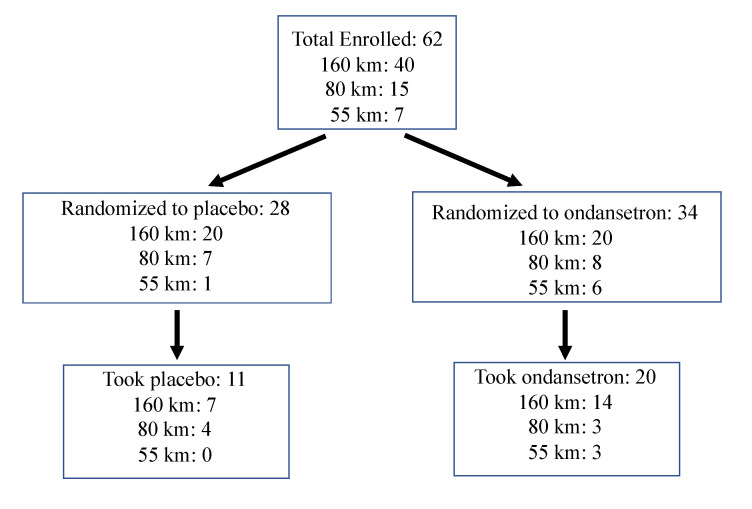
Diagram showing number of study participants and race distances from recruitment through intervention.

**Table 1 sports-09-00035-t001:** Comparison of the two study groups.

Characteristic or Variable	Ondansetron (*n* = 20)	Placebo (*n* = 11)	*p*-Value
Subject Characteristic			
Age (years)	48 ± 9	46 ± 8	0.62
Sex (% men)	60	27	0.14
Average weekly running distance (km) ^a^	80 ± 18	72 ±16	0.24
Highest weekly running distance (km) ^b^	112 (96–135)	120 (86–128)	0.88
Longest training run or race (km) ^b^	65 (50–100)	80 (50–100)	0.40
Seen physician for GI issues during running (%)	15	36	0.21
Frequently have nausea and vomiting during races (%)	80	73	0.68
Previously dropped out of ultramarathon (%)	55	27	0.26
During Race Characteristic			
Had vomiting during the race (%)	45	18	0.24
Distance treatment taken (km)	60 (42–79)	51 (32–64)	0.37
Severity of GI symptoms before treatment (points) ^c^	60 ± 27	60 ± 26	0.99
Took own medications during the race (%)	30	9	0.37
Received care at an aid station (%)	35	18	0.43
Unblinded during the race (%)	30	18	0.68
Finished race (%)	75	55	0.42
Treatment Outcome Variable			
Reported treatment benefit (%)	30	55	0.26
Treatment benefit (points) ^c^	31 ± 32	50 ± 42	0.15
Treatment benefit 1-h post-treatment (points) ^c^	35 (20–78)	40 (5–90)	0.90
Severity of GI symptoms at the finish (points) ^c^	34 ± 32	53 ± 43	0.21
Reported side effects from treatment (%)	15	36	0.21
Thought they took active medications (%)	63	73	0.70

Ordinal data are reported as mean ± SD or median (interquartile range) if skewed. Categorical data are reported as a percentage. ^a^ During the prior year. ^b^ During training for this race. ^c^ On 100-point scale. Abbreviation: GI, gastrointestinal.

## Data Availability

The data from this study are available upon request of the first author.

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
