# Peer review of "Treatment with Oral Ondansetron for Ultramarathon-Associated Nausea: The TOO FUN Study"

_sports, 2021, doi:10.3390/sports9030035_

Round 1

Reviewer 1 Report

Overview

The authors compared the effects of ondansetrone on GI distress on ultramarathoners. Runners in races of 55, 80, and 160 km length were recruited and then randomized to the placebo or medication and informed to use the medication if they experienced nausea or vomiting during the race. By chance, a higher percentage experiencing GI problems appeared to occur in the medication group (though p>0.05). For those having to use medication, there was a tendency of fewer percentagewise to report a benefit on ondansetrone (30) vs placebo (50) (again p>0.05). Although the study was preliminary and underpowered, the authors concluded that ondansetrone did not appear to be effective.

General Concerns

A number of statements could references for support. Specific below.

Design

If subjects were accrued over several races, how were external factors – environment, racecourse, sports drink on course, individual pre-race meals, and baseline GI state – accounted for? The temperature ranges provided seem variable.

Was there a performance difference between the 2 groups? Not sure how the investigators would standardize this given different race lengths. May be use the average speed of their race and weight it by the percentage of the total within a treatment for a given race.  The point is, if running intensity differed, that could be a confounding variable.

Specific Suggestions

Intro: Add pre-race anti-inflammatory meds to the list of potential causes of GI distress.

Methods, lines 77-80, sample size justification: Provide a bit more explanation as to why 30% was selected for placebo or treatment effect. As explained, the use of reference 6 as the pilot study does not make sense. A bit more description for the expected % for an effect would help. Was the pilot study done with ondansetrone? It appears to be descriptive, so how the bar was set for 30% effect of a placebo isn’t clear.

Results:

  • Any evidence that using ondansetron caused side effects or adverse reactions?
  • Did subjects who vomited keep ondansetron down? In other words, could the lack of an effect be due to the drug never getting into their system?

Discussion

Line 113 “Based on previous studies…” please provide references.

Lines 132-133: Provide a reference for the typical dosage.

Reviewer 2 Report

General comment

The study aimed to determine the effectiveness for Ondansetron to counteract exercise-associated

gastrointestinal distress during prolonged endurance running.  As highlighted by the authors, this question in the context of sport performance is poorly documented. Therefore, this article is concise and clearly presents the needs to investigate this question and the findings of this study. However, even in absence of a wide literature, I have mentioned some concerns that could deserve further details to better clarify the use and limits of this substance.

Introduction

L 44 – 45: Although of interest, I find the sentence relative to thermoregulation decontextualized from the paragraph and the previous sentence. May the authors should briefly add a specific part (even short) to present the knowledge about the adverse effects of Ondansetron on thermoregulation, cardiovascular control, or other factors related to performance during prolonged aerobic activities.

Materials and methods

L 60: What is the recommended dosing of this medication? I mean, how to control for the effectiveness of 1 capsule of Ondansetron between participants of different weight?

L 63: Given changes in gastrointestinal transit during prolonged running, how was decided the 4-hours time delay between 2 capsules intakes? Was it decided from your previous mentioned study or from manufacturer’s recommendation or something else? Please provide a reference whenever available. (I finally saw in the discussion section that this point is presented. Should the authors refer even partly to this consideration in the methods section as well).

L 73: Was the 1 to 100-point “effectiveness” scale specifically related to the quantification of the help provided by the treatment (e.g.; did for instance the criteria “Yes” quotes for 80 on the scale, and “Not sure” for 40), or did these two evaluations differ totally?

Reviewer 3 Report

This pilot study is important and of relevance to the running community and Sports community at large. Although important, there are some considerations I believe the authors should make. Overall, the comments and questions below are intended to improve the overall quality of the manuscript.

Title:

  • Should clearly state the primary findings of the study. Even though the results of the study are null, this is still relevant for anyone interested in reading this manuscript

Abstract:

  • Subject demographics would be helpful (sex, age, running habits)
  • Did you recruit subjects who often vomited during races, and the supplement did not help? This information has a great effect on how these data are interpreted

Introduction:

  • In the literature, are there any reported sex differences in the frequency and/or severity of nausea and vomiting during and endurance race.
  • Do environmental conditions (e.g., altitude, hot, cold, humid, etc.) influence frequency and/or severity of nausea and vomiting?

Methods:

  • Can ondansetron interact with other food/drink items that are commonly consumed food items (e.g., goos, electrolyte drinks, etc.) during an endurance race?
  • Methods should clearly state the demographics of the subjects included in the study.
  • It would be beneficial to the reader if this section was divided into subsections

Discussion:

  • Could there be a particular effect of ondansetron depending on the distance ran?
  • The study was not designed to determine cause and effect ondansetron, rather results are associated with ondansetron consumption. Wording should be adjusted accordingly throughout.
  • Could the habitual diet of the subjects have an effect on how they responded to ondansetron?
  • How might someone design a study that would determine cause and effect of ondansetron on nausea and vomiting during an endurance race?

Round 2

Reviewer 3 Report

Thank you for addressing my comments. I believe the manuscript is significantly improved.

Author Response

Thank you.